# Lipid Saturation and the Rheology of Human Tear Lipids

**DOI:** 10.3390/ijms20143431

**Published:** 2019-07-12

**Authors:** Georgi As. Georgiev, Douglas Borchman, Petar Eftimov, Norihiko Yokoi

**Affiliations:** 1Biomaterials and Biointerfaces Laboratory, Department of Optics and Spectroscopy, Faculty of Physics, St. Kliment Ohridski University of Sofia, Sofia 1164, Bulgaria; 2Department of Ophthalmology and Visual Sciences, University of Louisville, Louisville, KY 40202, USA; 3Department of Cytology, Histology and Embryology, Faculty of Biology, St. Kliment Ohridski University of Sofia, Sofia 1164, Bulgaria; 4Department of Ophthalmology, Kyoto Prefectural University of Medicine, Kyoto 602-8566, Japan

**Keywords:** human meibum, acyl chain saturation, Langmuir films, tear film stability, surface properties

## Abstract

Elevated levels of acyl chain saturation of meibomian lipids are associated with enhanced tear film (TF) stability in infants to shortened TF breakup time with meibomian gland dysfunction. Thus, the effect of saturation on the surface properties of human TF lipids (TFLs) using a Langmuir surface balance and Brewster angle microscopy was studied. Lipid phase transitions were measured using infrared spectroscopy. The raise in the % of saturation resulted in thicker, and more elastic films at π = 12 mN/m, with the effects being proportional to the saturation level. At the same time, at lower (≤10 mN/m) π, the raise in saturation resulted in an altered spreading and modified structure of TFL layers. The strong impact of saturation on TFL surface properties correlated with a saturation induced increase of the TFL acyl chain order, phase transition temperature, and lipid–lipid interactions. The native TFL order and π_max_ were significantly greater, compared with native meibum collected from the same individual. Aggregation of lipids on the tear surface due to saturation was not as significant as it was for meibum. Although the surface pressure/area isotherms for TFL were similar for meibum, differences in rheology and phase transition parameters warrant the study of both.

## 1. Introduction

A thin film of lipids, called the tear film lipid layer (TFLL), covers the surface of the tear film (TF) and serves important functions. The TFLL ensures that the TF has a low surface tension, and an immobile air/TF interface that opposes the outflow of aqueous tears in an open eye [1,2,3,4,5]. The TFLL is also antibacterial and servers to dam, lubricate, allow for proper refraction, and suppresses exposure to UV rays [6]. The major source of tear lipids (TLs) are the meibomian glands [6], but other sources may include lipocalin bound phospholipid (citations in reviews [2,7] and others [8,9,10,11,12,13,14]) or by eyelid sebum [15,16,17]. The composition (citations in reviews [2,7] and others [8,9,10,11,12,13]) and physical properties [18] of TLs are slightly different compared with meibum lipids.

Relationships between TFLL composition, structure, and function could provide insight into the etiology of dry eye. One measurement of lipid structure is termed ‘order,’ which is indirectly related to lipid fluidity that has an additional dynamic component. It is interesting that tear film stability and meibum lipid hydrocarbon order both increase in the order: Meibum from donors without dry eye between the ages of 0 and 25 years old [18,19] < meibum from donors with meibomian gland dysfunction [18,20,21] < meibum donors who are susceptible to severe dry eye who have had hematopoietic stem cell transplantation [18,22,23]. When dry eye signs and symptoms were ameliorated with treatment, lipid order was restored [21]. Correlation does not necessitate cause, but the relationship between hydrocarbon chain order and stability is intriguing and at best, lipid order is a marker for age and dry eye with a positive correlation of 93% [24,25]. In vitro, more ordered lipids result in the formation of stiffer, thicker, and more elastic films at high surface pressures. However, at low surface pressure, occurring at high film areas resembling the open eye condition, the increase in saturation impaired the spreading of human meibum and more heterogeneous layers containing thick ‘islands’ of aggregated lipids were observed [26]. One may speculate that this latter effect would also result in suppressed spreading of meibum over the aqueous tear layer at the ocular surface.

The phase transition temperature strongly correlates with lipid order. Lipid unsaturation, the topic of the current study, is the major factor that contributes to lipid disorder and the phase transition temperature [26,27,28]. Other factors that may contribute to lipid order include proteins [25], hydrocarbon chain branching [29], cholesterol [30], cholesteryl esters [31], and hydrocarbon chain length.

As the composition and physical properties of TLs are different than that of meibum (above) and saturation contributes to the physical properties of meibum [26,27,28,32] we extracted TLs from tears on Schirmer’s strips and catalytically saturated the TLs to determine if saturation contributes to the physical properties of TLs as it does to meibum. The physical and thermodynamic properties of TLs with a range of saturation levels were measured using infrared spectroscopy and their rheology determined by Langmuir trough technology.

## 2. Results

### 2.1. H-NMR Spectroscopy

The ^1^H NMR spectrum of human tear lipid was typical of that collected on a 700 mHz spectrometer (Figure 1). Band assignments were made from ^1^H and ^13^C NMR studies [15,33]. The largest resonance in this region was observed at 5.32 ppm, with a shoulder at 5.35 ppm assigned to protons of the *cis* =CH moieties from hydrocarbon chains and to the proton attached to carbon #6 of cholesterol esters, respectively. The resonance at 4.6 ppm is from cholesteryl esters and the resonance at 4.0 ppm is from wax esters (Figure 1). The resonance near 5.1 ppm is assigned to squalene [15]. Spectra of the catalytically hydrogenated sample did not have a detectable *cis* =CH resonance at 5.32 ppm, indicating that it was completely saturated (Figure 1b).

### 2.2. Effect of Saturation on Human Tear Lipid Phase Transition Parameters.

Infrared spectroscopy was used to measure lipid–lipid interactions and composition. The CH_2_ stretching and bending bands are predominant in the infrared spectra of lipids. The CH stretching region of meibum is composed of six major bands (Figure 2) [34]. Note the catalytically hydrogenated sample has no =CH stretching moieties (Figure 2, bottom). In this study, we used the ${\tilde{v}}_{\mathrm{sym}}$ near 2850 cm^−1^ to estimate the *trans* to *gauche* rotamer content of the hydrocarbon chains. The ${\tilde{v}}_{\mathrm{sym}}$ increased with an increase in temperature and the number of *gauche* rotamers, concurrent with a decrease in intensity (Figure 3).

The maximum ${\tilde{v}}_{\mathrm{sym}}$ and phase transition Δ entropy and Δ enthalpy (Section 4.5) were significantly higher and the lipid order at 33.4 °C was significantly lower for TLs extracted from the tips compared with TLs extracted from the stems, *p* = 0.006, *p* < 0.001, *p* < 0.001, *p* < 0.001, respectively (Table 1). TLs from the tips and stems were combined, catalytically hydrogenated, and studied using FTIR and Langmuir trough technology.

The lipid phase transition temperature for TLs increased with saturation, as expected, from about 30 °C to 51 °C (Figure 4b). Lipid order was measured close to the surface temperature of the human eye, 33.4 °C, by extrapolating the ${\tilde{v}}_{\mathrm{sym}}$ at 33.4 °C from the fit of the phase transition and then converting ${\tilde{v}}_{\mathrm{sym}}$ to the percentage of *trans* rotamers. Lipid order at 33.4 °C increased from about 30% to 80% between 0% and 25% saturation (Figure 4a). Above 25% saturation, the lipid order was at the maximum level of 80%. Lipid cooperativity (Figure 4e), Δ enthalpy (Figure 4c), and Δ entropy (Figure 4d) increased substantially (*p* < 0.0001) with saturation, r = 0.971, 0.969, 9.87, respectively. The order of TLs was significantly 35% larger, 42 ± 2% *trans* compared with meibum collected from the same individual, 31 ± 1% *trans*.

### 2.3. Effect of Catalytic Saturation on Human Tear Lipid Rheology

Saturation sequentially increased the maximum surface pressure (at 10% surface area), indicating increased molecular packing density achieved at completion compression (Figure 5). Saturation changed the shape of the reciprocal compressibility modulus curves, which indicates it caused the formation of a more condensed and solid surface phase at π > 10 mN/m (Figure 6). More condensed solid surface phases (white) at lower surface pressures below 8 mN/m are visible in the Brewster angle microscopy micrographs (Figure 6b,c). An increase in catalytic saturation shifted the surface pressure relaxation transients to higher surface pressures (a manifestation of increased elasticity) and changes the shape of the transients (i.e., there is change the molecular reorganization processes in the layer reflecting the alterations in the phase coexistence in the layers) (Figure 7). The 1% to 4% saturated samples’ transients (as well as their surface pressure/area isotherms) were identical to the one of native tear lipid. The other transients were subjected to Fourier transformation analysis and the results are summarized in Figure 8. The curves in Figure 8 indicate that saturation causes an increase in the real (= elastic) part (E^R^) of the complex dilatational modulus for the entire frequency range and a decrease in the tangent of the phase angle. The latter indicates an increase in the contribution of the elastic modulus vs. the contribution of the viscous modulus with saturation. These changes with saturation indicate that the phase coexistence and organization of tear lipid films changes with saturation. Cole–Cole plots confirm that there are at least two processes contributing to the relaxation transients. The Maxwell rheological model equations are summarized in Table 2.

In summary, an increase in saturation increases the elasticity and maximum surface pressure and changes the surface phases and phase transitions of the tear lipid layers. An increase in saturation >50% elevates the tear film’s elasticity, but modifies the spreading and results in more heterogeneous layers over a broad surface pressure range. Saturation does not significantly affect the surface properties of films formed by human tears below a surface pressure of 8 mN/m.

## 3. Discussion

One of the major findings of our study is that TL saturation, up to 20% that of native meibum, increased the phase transition temperature and order of the hydrocarbon chains. The correlation between the hydrocarbon order and phase transition temperature has been firmly established [26,27,28,32]. As saturation eliminates ‘kinks’ in the hydrocarbon chains due to the *cis* conformation of the double bond, the chains become straighter, allowing for more contact between hydrocarbon chains and greater Van der Waals interactions. Saturation induced stronger lipid–lipid interactions are evident in a linear increase in the change in phase transition enthalpy and entropy with saturation. Saturation also increases the cooperativity of the phase transition. This may be because the system becomes more homogeneous with less of a difference between saturated and unsaturated moieties. 

Differences were found between the lipid phase transition parameters of lipids extracted from the tips of Schirmer’s strips and the stems. The differences may be due to tear lipids bound to soluble proteins that migrate up the Schirmer’s strip. Differences between TL extracted from tips and stems of Schirmer’s strips have been characterized [12]. Stronger lipid–lipid interactions as discussed above influence the rheology of TL reflected in an increase in the elasticity and maximum surface pressure and changes in the surface phases [26,27,28]. Tear lipids need to be ordered enough on the surface of the tear film to withstand the sheer force of a blink and resist tear film breakup, a process in which lipid–lipid interactions are broken. The lipids cannot be too ordered or they will aggregate into ‘islands’ (lateral phase separation) and not spread [2,26]. Therefore, the balance between saturation and unsaturation needs to be carefully balanced in vivo. Saturation does not affect the interfacial properties of surface films formed by human tears, as below 8 mN/m, the surface film is rich in non-lipid/non-meibomian compounds that come from aqueous tears and are essentially unaffected by the catalytic saturation. These aqueous tear compounds acted as spreading agents for all tested saturation levels. One may speculate that phospholipids found almost exclusively in TL scan form a monolayer at the interface between the TFLL and tear aqueous layer. As phospholipids are amphipathic, their hydrophilic head groups are expected to face the aqueous layer and their hydrophobic tails are expected to face the bulk lipid layer above, serving as a scaffolding for the hydrophobic wax and cholesteryl lipid bulk layer.

It is interesting that a small 1% to 4% increase in saturation did not significantly influence TL rheology, which highlights the resiliency of the TFLL.

### Meibum Verses TL

As indicated, the majority of the lipids in tears are from meibum, but there are significant compositional differences between tear lipids and meibum lipids that could influence the structure, rheology, and function of the tear film lipid layer. Saturation affected tear lipids more significantly than meibum lipids [26] as the slopes of the linear regression lines in the curves of saturation verses ΔH, ΔS, and cooperativity were steeper for TLs. This difference between tear lipids and meibum lipids is significant for native lipids and at higher non-physiological levels of saturation (>70% saturation). It is interesting that the order of native TLs was 35% greater than native meibum collected from the same individual. This has been observed in another study [35]. The compositional moiety(s) that cause the difference is unknown at present. The current study suggests saturation could play a significant role, however, other factors, such as chain length, hydrocarbon chain branching, and amphipathic compounds, could contribute to the difference.

The phase transition temperature and order of tear lipids that were catalytically saturated was not significantly different at the same level of saturation compared with meibum lipids that were catalytically saturated. Therefore, the differences in rheology between tear and meibum lipids at a given saturation level are not due to the strength of lipid–lipid interactions, but rather other factors, such as the non-lipid/non-meibomian compounds that come from aqueous tears that are unaffected by catalytic saturation.

As the native TL order was significantly 35% greater than native meibum collected from the same individual, one would expect from the current study that π_max_ would be greater for TLs compared with ML as observed. The film surface compressional modulus, Cs^−1^, at a given surface pressure occurred over a smaller range, 11 to 12 mN/m, compared with 8 to 80 mN/m for meibum [26]. The inflexion points in the π/Cs^−1^ dependencies indicate the surface pressures at which significant reorganization of the surface film takes place in the course of the film compression. Thus, TL lipids reorganize over a tighter range of surface pressure compared with meibum [26]. There was no noticeable change in the lift off area of TLs with saturation, whereas for meibum, the lift off area increased above 50% saturation, which indicates that saturation impaired the spreading of meibum but not TLs. As discussed above, aqueous tear compounds, perhaps amphipathic compounds in TLs not found in meibum, act as spreading agents at all tested saturation levels.

Among the many factors discussed in the Introduction, saturation could contribute to or be a marker for differences in lipid order and tear film stability, especially between the ages 0 and 25 years and among individuals who are susceptible to severe dry eye who have had hematopoietic stem cell transplantation, as elevated saturation levels for both cohorts have been documented. Indeed, saturation plays a significant role in ordering lens, retinal, muscle, and a variety of synthetic lipids [21,22,23,27,28,32].

In conclusion, saturation increased TLs order, phase transition temperature, lipid–lipid interactions, elasticity, and maximum surface pressure and changed surface phases. Lipid order and π_max_ were greater in TLs than meibum collected from the same individual. Aggregation of lipids on the tear surface due to saturation was not as significant as it was for meibum. Although the surface pressure/area isotherms for TLs were similar to ML, differences in rheology and phase transition parameters warrant the study of both.

## 4. Methods

The materials, diagnosis of normal status, collection and extraction of lipid from meibum, and catalytic hydrogenation sections were identical and copied from [28].

### 4.1. Materials

Silver chloride windows for infrared spectroscopy were obtained from Crystran Limited, Poole, United Kingdom. Platinum (IV) oxide was obtained from the Sigma Chemical Company (Sigma Chemical Company, St. Louis, MO, USA). Unmarked Schirmer’s strips without ink ruling were purchased from Alcon Laboratories (Alcon Laboratories, Fort Worth, TX, USA).

### 4.2. Collection and Extraction of Tear Lipids

Tears were collected on 54 Schirmer’s strips from each eye of a 63-year-old Caucasian male in the morning for 1 month. The donor status was normal as his meibomian gland orifices showed no evidence of keratinization or plugging with turbid or thickened secretions and no dilated blood vessels were observed on the eyelid margin. The donor did not recall having dry eye symptoms and did not wear contact lenses. Written informed consent was obtained from all donors. Protocols and procedures were reviewed by the University of Louisville Institutional Review Board as well as the Robley Rex Veterans Affairs Institutional Review Board. All procedures were in accordance with the Declaration of Helsinki. Protocols and procedures for the current retrospective study were approved by the University of Louisville Institutional Review Board (# 11.0319, August 2016). The leading tip (5 mm) of the Schirmer’s strip was bent at a 90° angle and subsequently placed over the edge of the lower eyelid for 5 min. The lower portion (5 mm) of the strips was cut off and the upper and lower portions were each placed in a vial filled with argon to prevent oxidation. Care was taken not to contaminate the lower end of the test strip with finger lipids.

Lipids were extracted from the two pools of 54 Schirmer’s strips that were placed in the two glass vials using 5 mL of methanol that had been bubbled with argon for 5 min. The strips and methanol were sonicated with a Sonifier1 cell disrupter microprobe (Branson Ultrasonics, Danbury, CT, USA) three times for 15 s each with a 2 min cooling period between sonication. The methanol was decanted and centrifuged at 10,000 rpm for 15 min to remove methanol insoluble impurities. The methanol was decanted again with care not to disturb the pellet. The steps above were repeated using CHCl_3_ and then again with benzene instead of methanol. The CHCl_3_ and benzene lipid extracts were added to the methanol lipid extract and the solvents were evaporated under a stream of argon. CDCl_3_ (1 mL) was added to each sample for NMR analysis and catalytic hydrogenation.

### 4.3. Catalytic Hydrogenation

Half of each of the pooled tear lipid samples was decanted to be catalytically hydrogenated. Saturated tear lipid was prepared as for sphingomyelin [36,37]. The samples were reduced catalytically with hydrogen over platinum (IV) oxide (7.4 mg) at room temperature and atmospheric pressure for approximately 4 h with stirring. The catalyst was separated from the solution by centrifugation. The solution was decanted with care not to disturb the pellet of catalyst. Catalytically saturated samples were quantitatively mixed with sample that was not catalytically saturated to provide mixtures containing 1%, 2%, 3%, 4%, 5%, 10%, 25%, 50%, and 67% catalytically saturated meibum.

### 4.4. NMR Spectroscopy

Spectral data were acquired using a Varian VNMR 700 MHz NMR spectrometer (Varian, Lexington, MA, USA) equipped with a 5 mm ^1^H{^13^C/^15^N} ^13^C-enhanced cold probe (Varian, Palo Alto, CA, USA). Spectra were acquired with a minimum of 250 scans, 45° pulse width, and a relaxation delay of 1.000 s. All spectra were obtained at 25 °C. The TMS resonance was set to 0 ppm. Commercial software (GRAMS 386; Galactic Industries Corp., Salem, NH, USA) was used for spectral deconvolution and curve fitting. 

### 4.5. Measurement of Lipid Phase Transitions Using Fourier Transform Infrared Spectroscopy

Lipid phase transitions were measured as described previously [29]. Approximately 500 µL of sample in CDCl_3_ was applied to a AgCl infrared window. The solvent was evaporated under a stream of argon gas and the window was placed in a lyophilizer for 4 h to remove all traces of solvent. Infrared spectra were measured using a Fourier transform infrared spectrometer, Nicolet 5000 Magna Series (Thermo Fisher Scientific, Inc., Waltham MA, USA). Lipid on the AgCl window was placed in a temperature-controlled infrared cell. The cell was jacketed by an insulated water coil connected to a circulating water bath, modelR-134A (Neslab Instruments, Newton, NH, USA). The sample temperature was measured and controlled by a thermistor touching the sample cell window. The water bath unit was programmed to measure the temperature at the thermistor and to adjust the bath temperature so that the sample temperature could be set to the desired value. The rate of heating or cooling (1 °C/15 min) of the sample was also adjusted by the water bath unit. Temperatures were maintained within ±0.01 °C. Exactly 100 interferograms were recorded and averaged. Spectral resolution was set to 1.0 cm^−1^.

The frequency of the symmetric CH_2_ stretching band near 2850 cm^−1^ (${\tilde{v}}_{\mathrm{sym}}$) was used to estimate the content of *trans* and *gauche* rotamers (lipid order) in the hydrocarbon chains as described [18]. Lipid phase transitions were quantified by fitting the data to a two-state, sigmoidal equation using Sigma plot 10 software (Systat Software, Inc., Chicago IL, USA) as follows:$${\tilde{v}}_{\mathrm{sym}}=({\tilde{v}}_{\mathrm{sym}})\mathrm{minimum}+(({\tilde{v}}_{\mathrm{sym}})\mathrm{maximum}-({\tilde{v}}_{\mathrm{sym}})\mathrm{minimum})/(1+{\left(\mathrm{temperature}/\mathrm{Tc}\right)}^{\mathrm{hillslope}}),$$
where ${\tilde{v}}_{\mathrm{sym}}$ is the frequency of the symmetric CH_2_ stretching band near 2850 cm^−1^ and Tc is the phase transition temperature. Hillslope is the relative cooperativity. The broader the phase transition, the smaller the value for the hillslope. This term describes how the order of a lipid influences that of neighboring lipids. The data for the percentage of *trans* rotamer were used to calculate the phase-transition enthalpy and entropy from the slopes of Arrhenius plots.

### 4.6. Compression Isotherms

Surface pressure–area (π-A) isotherms were measured using Langmuir surface balance µTrough XS, area 135 cm^2^, volume 100 mL (Kibron, Helsinki, Finland) by the Wilhelmy wire probe method (instrumental accuracy 0.01 mN/m) [38,39,40]. The trough subphase was physiological saline solution buffer (PBS, pH 7.4). Human MGS/tears sample dissolved in chloroform were deposited (35 µL of 1 mg/mL) over the air/saline surface with a microsyringe (Hamilton Co., Reno, NV, USA). The trough was positioned under an acrylic cover to protect the surface from dust and to suppress the evaporation of the saline solution subphase. After 15 min for chloroform evaporation, film compression was performed by two symmetrically moving barriers. Dynamic compression–expansion isocycling of the layer area was done with the maximum barrier’s rate (70 mm/min) at which there was no film leakage. Ten consecutive cycles were performed with each film studied. Normally, after the third cycle, the shape of the π(A) curves remained constant and those π(A) isotherms were presented and analyzed. All isotherms were repeated at least three times; the difference between the repetitions was less than 2%. The π(A) hysteresis was minimal between repeated isocycles of meibum films and that is why only compression isotherms are presented. The experiments were done at 35 °C. The films’ morphology was monitored by MicroBAM, KSV-NIMA, Brewster angle microscopy (Quantum Design GmbH, Darmstadt, Germany). The film surface compressional modulus, Cs^−1^, at a given surface pressure was calculated from the π/A compression isotherm as described [41,42].

### 4.7. Stress–Relaxation Studies via the Small Deformations Method

In order to gather information about the dilatational viscoelasticity of meibum/tear films that were catalytically saturated, the relaxation of the surface pressure was monitored after a small rapid compression deformation was applied to the surface film as described in detail in [42,43,44]. Firstly, the film was compressed to the initial surface pressure, π_0_, of 15 mN/m. Then, the lipid film was instantaneously and slightly contracted with a compression step, ∆*A*/*A*o = 5 ± 1% (*A*o is initial film area, and ∆*A*—area change). As discussed elsewhere [40,41,42,43,44], no assumptions were made about the surface film structure or the physical nature of the relaxation processes (e.g., diffusion to/from the bulk solution, molecular rearrangements, exchange with secondary adsorption layers, etc.).

The Fourier analysis of the relaxation transients was performed as previously described [40,41,42,43,44] utilizing commercial Fourier transform software provided by Kibron Inc. (Helsinki, Finland). The data were subjected to further analysis by constructing their corresponding Cole–Cole lots (i.e., graph of E_IM_ vs. E_R_) as described [40,41,42,43,44]. In the context of stress relaxation, the number of peaks in the Cole–Cole plot indicates the number of relaxation processes contributing to the relaxation of π [40,41,42,43,44].

### 4.8. Statistics

Curves were fitted using Sigma plot 10 software (Systat Software, Inc., Chicago IL, USA) and the confidence levels, *p*, were obtained from a critical value table of the Pearson product-moment correlation coefficient. A value of *p* < 0.05 was considered statistically significant.

## Figures and Tables

**Figure 1 ijms-20-03431-f001:**
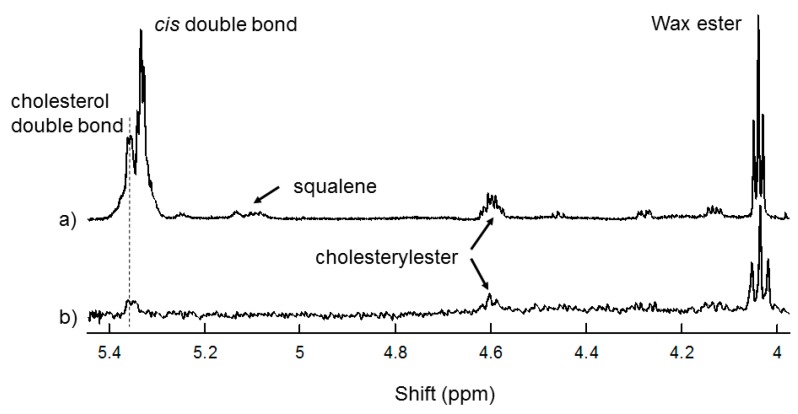
^1^H-NMR spectrum of tear lipids from (**a**) a 63-year-old Caucasian male with no dry eye. (**b**) ^1^H-NMR spectrum of tear lipids in Figure (**a**) that were catalytically saturated. Note that there are no *cis* hydrocarbon chain double bonds in the saturated sample spectra.

**Figure 2 ijms-20-03431-f002:**
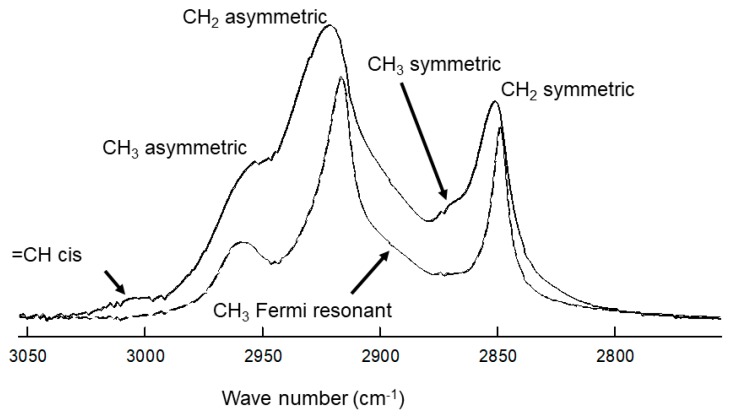
Infrared CH stretching region spectrum of tear lipids from (top) a 63-year-old Caucasian male with no dry eye. (bottom, ---) Infrared spectrum of tear lipids that were catalytically saturated. Note that there are no *cis* hydrocarbon chain double bonds in the saturated sample spectra.

**Figure 3 ijms-20-03431-f003:**
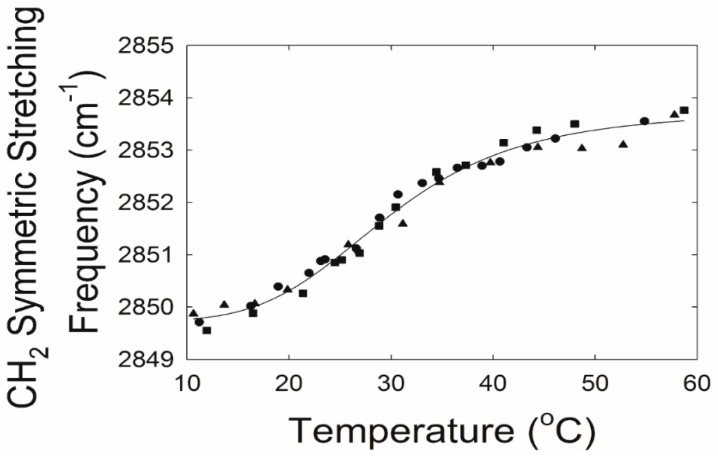
Lipid phase transitions of tear lipids from a 63-year-old Caucasian male with no dry eye. The CH_2_ symmetric stretching frequency is related to the lipid structural order. The higher the value for the frequency, the more disordered the lipid. Symbols are different trials. Solid line is the fit of the data to a sigmoidal equation (see Section 4).

**Figure 4 ijms-20-03431-f004:**
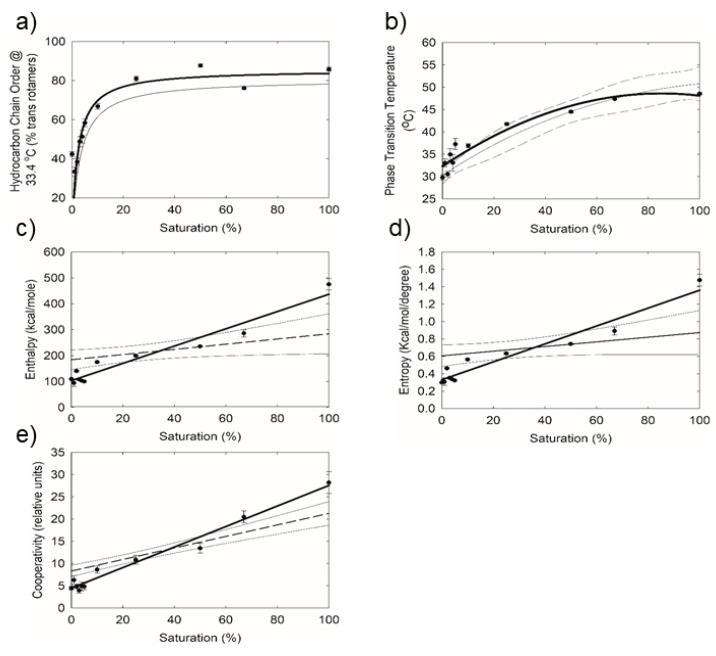
Lipid phase transition parameters for human tear lipid that was mixed with catalytically hydrogenated tear lipid from the same pool: (**a**) hydrocarbon chain order at the tear film surface temperature, (**b**) Phase transition temperature, (**c**) change in enthalpy, (**d**) change in entropy, (**e**) cooperativity. The % saturation is the percentage of catalytically saturated tear lipid mixed with normal tear lipid. Data are the average ± the standard error of the mean. (---) Data from a study of meibum lipid for comparison (from [26]). (light large dashed) 95% confidence interval of meibum lipid data (from [26]).

**Figure 5 ijms-20-03431-f005:**
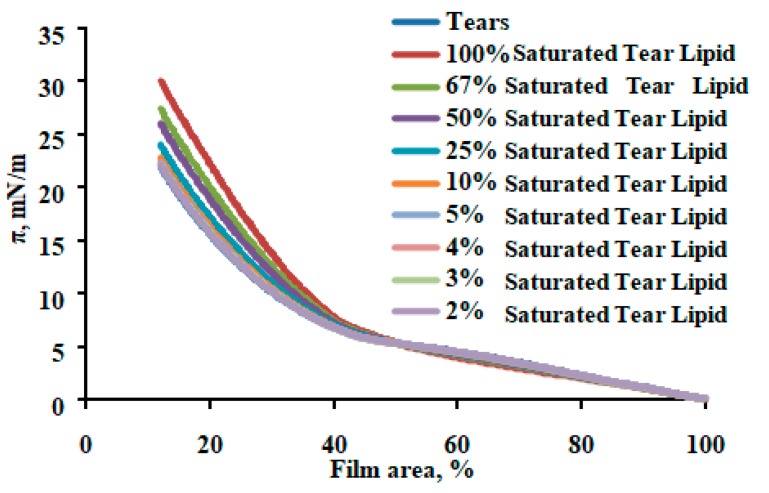
In the case of tears, catalytic saturation enhances maximum surface pressure without alteration of spreading (= the low π part of the isotherms). This reflects two points: (i) the tear film lipid layer (TFLL) is somewhat different from meibum and (ii) TFLL tolerates saturation and its properties are enhanced by it.

**Figure 6 ijms-20-03431-f006:**
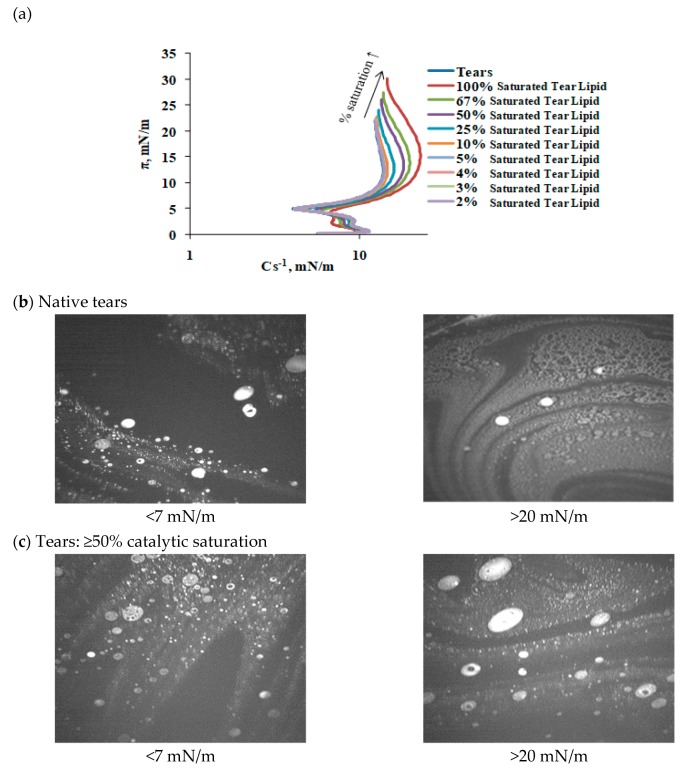
The increase in saturation shifts the reciprocal compressibility modulus to higher values at higher surface pressures (**a**), however, at low π values, the curves are almost identical. This suggests that the spreading and phase coexistence in tear layers are tolerant to catalytic saturation. This is well illustrated by the Brewster angle microscopy (BAM) micrographs (**b**,**c**).

**Figure 7 ijms-20-03431-f007:**
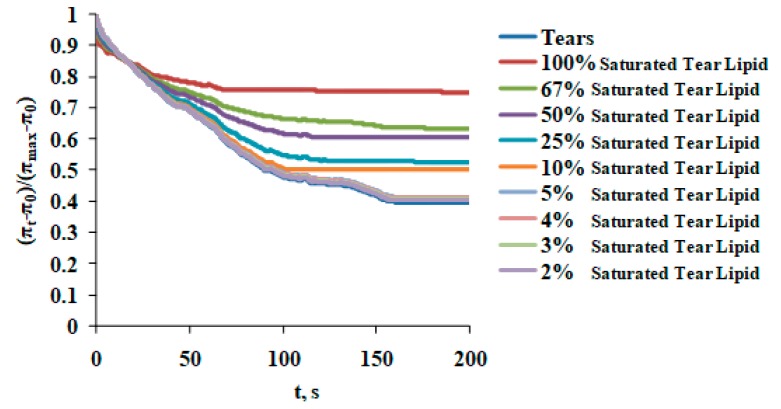
Increase in catalytic saturation shifts the surface pressure relaxation transients to higher surface pressures (a manifestation of increased elasticity). The shape of the transients is complex and reflects the composite composition of the tear layers. The 2% to 5% saturated samples’ transients are identical to the one of native tears. The rest of the transients were subjected to Fourier transformation analysis and the results are summarized in Figure 8.

**Figure 8 ijms-20-03431-f008:**
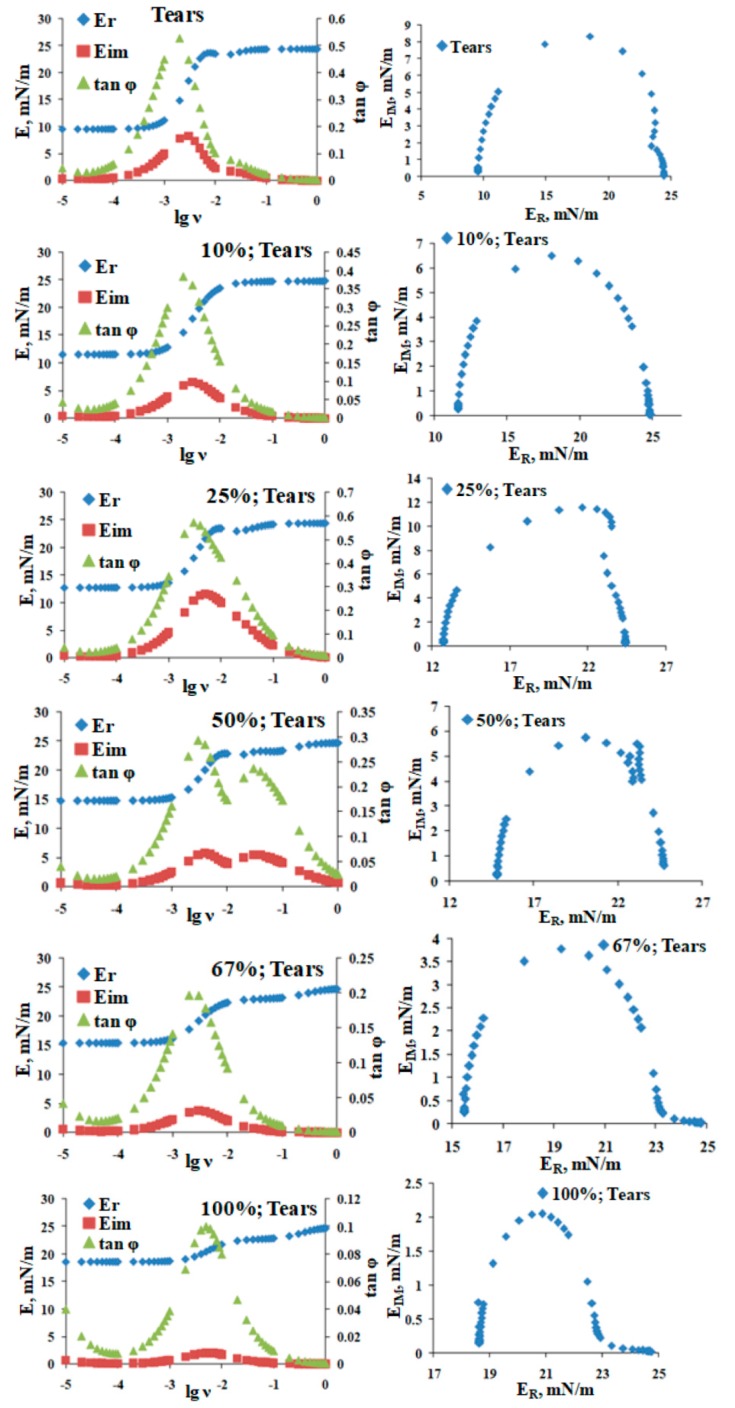
Rheological parameters obtained by Fourier transformation of the relaxation transients from Figure 7. Right panels denote the frequency dependences of the real and imaginary part of the complex modulus and of the tangent of the phase angle; the left panels show Cole–Cole plots of E_IM_ vs. E_R_. The % denotes the degree of tear lipid saturation.

**Table 1 ijms-20-03431-t001:** Phase transition parameters of tear lipids extracted from the tips and stems of Schirmer’s strips from a 63-year-old male.

Table Cont.	Tear Lipid from Tips	Tear Lipid from Stems	***p***
**Minimum (cm^−1^)**	2849.74 ± 0.09	2849.5 ± 0.2	0.020
**Maximum (cm^−1^)**	2853.74 ± 0.13	2854.6 ± 0.3	0.013
**Tm (°C)**	29.8 ± 0.6	28.4 ± 1.0	0.24
**Hill (cooperativity)**	−4.5 ± 0.4	−4.6 ± 0.8	0.91
**Order 33.4 °C**	42.4 ± 1.3	32.9 ± 1.3	>0.0001
**Δ Entropy (Kcal/mol)**	0.298 ± 0.004	0.452 ± 0.007	>0.0001
**Δ Enthalpy (Kcal/mol/degree)**	90 ± 1	136 ± 2	>0.0001

**Table 2 ijms-20-03431-t002:** Maxwell rheological model equations for the tear layer samples. All the equations fit the raw transient at Figure 7 with R^2^ ≥ 0.98.

Composition	Maxwell Rheological Model Equation
Tears	Δπ = 1.2 × 10^−3^exp(–t/1) + 0.65exp(–t/73) + 0.337
100%; Saturated Tear Lipid	Δπ = 7.1 × 10^−2^exp(–t/0.98) + 0.162exp(–t/30) + 0.75
67%; Saturated Tear Lipid	Δπ = 5.7 × 10^−2^exp(–t/1) + 0.162exp(–t/52) + 0.62
50%; Saturated Tear Lipid	Δπ = 2.9×10^−2^exp(–t/0.1) + 0.37exp(–t/43) + 0.596
25%; Saturated Tear Lipid	Δπ = 2.3×10^−2^exp(–t/0.1) + 0.48exp(–t/53) + 0.499
10%; Saturated Tear Lipid	Δπ = 1.9×10^−3^exp(–t/0.79) + 0.52exp(–t/47) + 0.47

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
