# Peer review of "Lipid Saturation and the Rheology of Human Tear Lipids"

_ijms, 2019, doi:10.3390/ijms20143431_

Reviewer 1 Report

see attached file

Author Response

We thank the reviewers for their time and expertise in improving the manuscript. We have addressed all of their comments that appear in red in the revised draft. We hope that you find the manuscript suitable for publication.

Reviewer 1

1. I have no major issues with this manuscript but I should highlight that the

techniques used are outside my area of expertise. I do have several minor points.

Using the terminology “2%, tears; 3%, tears” etc. in table 2 and figures 5 and 7 is

confusing. Having tears after the comma is redundant. Why not replace with

saturation, e.g. 2% saturation. I think this would make the table and figures

easier to follow.

We replaced % tears with % saturated tear lipids

2. In the conversion to pdf it appears that p has be converted to 6 throughout the

discussion.

This was beyond our control. We will make sure all of the conversions are correct in the proofs should the manuscript be accepted for publication.

3. Minor typos/suggestions:

Line 52: lipids, which

Line 56: define HSCT

Lines 94-104: several spaces between words missing

Line 96 and figure 2: there is no b in the figure, please indicate which spectrum is

which.

Line 114: In the figure 4 caption, should you be referring to Figure 3?

Line 120: Use the same nomenclature in the text and Table 1. Also show

significance in table 1.

Line 168: There is no image for Figure 6 a).

Line 207: Tear lipids need to be ordered enough …

Lines 229-230: native TFL was 35% greater than native meibum…

Line 236: between tear and meibum lipids ….

Line 240: As native TL order was 35% greater than native meibum ….

Lines 257-258: Lipid order and pmax were greater in TL than meibum collected

…..

Line 260: similar to ML …

Line 288: repeated using CHCl3

All of the above typos and suggestions were addressed in the revised manuscript. We double checked the draft we submitted and found that between submission and the draft you received, the figures were placed in the wrong places and many of the symbols were not converted correctly.

Reviewer 2

3. the references are not sequentially numbered, redundant statements are made, figure legends are incomplete, etc.

The references are now sequentially numbered

4. Lines 51-52- This is another trivial statement that could be made on science outreach activities to non-specialized audience, but not in a scientific paper, I mean compare order and fluidity with butter and olive oil.

Sorry. We should always be aware of the audience. Most of the readers of our publications in Ophthalmic journals are clinicians and we needed to explain many of the scientific concepts in simple terms. As most of the readers of IJMS will be scientists, we eliminated many of these simplistic statements.

5.  Line 56 – Define HSCT. This can be immediately found upon search, but should be described in the text.

 Thank you. HSCT was spelled out in the two places it was used.

6.    Line 79 – Average NMR spectra were typical of that collected on a 700 MHz machine???

The sentence was correct to: The 1H NMR spectrum of human tear lipid was typical of that collected on a 700 mHz spectrometer (Figure 1). 

7. Lines 92/93- Trivial- There are a large number on CH2 groups in the lipid, so these are the major NMR resonances…

We agree that this sentence is trivial and was removed.

 8. Line 99/100 - Although mentioned, the sigmoidal equation should be described.

The equation is now described in the Methods.

9.  Lines 99/100 – The transition is not fitted to the equation. Equations are fitted to the data.

Thank you. The sentence was corrected to:

Lipid phase transitions were quantified by fitting the data to a two-state, sigmoidal equation using Sigma plot 10 software (Systat Software, Inc., Chicago IL) as follows:

10. Line 114- Complete Figure Legend.

Thank you. The revised legend is:

Figure 3. Lipid phase transitions of tear lipids from a 63 year-old Caucasian male with no dry eye. The CH2 symmetric stretching frequency is related to lipid structural order. The higher the value for the frequency the more disordered the lipid. Symbols are different trials. Solid line is the fit of the data to a sigmoidal equation (See Methods).

 11. Line 120 – Describe how the thermodynamic data was obtained.

A paragraph was added to section 4.5.

Reviewer 2 Report

This work addresses a relevant biological system, the meibomian lipids. For this purpose infrared spectroscopy and optical microscopy were used, and it should be stressed that the work related to the rheological properties, as determined from Langmuir trough methodologies, is carried out at state-of-the-art level, namely the Fourier transform analysis.

Although there are nonsignificant flaws in the work, this should underwent strong major alterations, since the manuscript is presented in a non-careful and by far non-acceptable way. As examples of this, the references are not sequentially numbered, redundant statements are made, figure legends are incomplete, etc.

In addition, there is a very high number of self-citations.

I will describe below just a few examples. This is not an exhaustive description:

i)                    Although this is not determined in the work (usual methodologies are NMR and transient-state fluorescence anisotropy), order parameter is the quantitative approach.

Certainly that saturated chains have less kinks in the structures, so order (determined by Van der Waals forces/packing), are greater. There are several trivial statements about this.

ii)                   Lines 51-52- This is another trivial statement that could be made on science outreach activities to non-specialized audience, but not in a scientific paper, I mean compare order and fluidity with butter and olive oil.

iii)                 Line 56 – Define HSCT. This can be immediately found upon search, but should be described in the text.

iv)                 Line 79 – Average NMR spectra were typical of that collected on a 700 MHz machine???

v)                  Lines 92/93- Trivial- There are a large number on CH2 groups in the lipid, so these are the major NMR resonances…

vi)                 Line 99/100 - Although mentioned, the sigmoidal equation should be described.

vii)               Lines 99/100 – The transition is not fitted to the equation. Equations are fitted to the data.

viii)              Line 114- Complete Figure Legend.

ix)                 Line 120 – Describe how the thermodynamic data was obtained.

Author Response

We thank the reviewers for their time and expertise in improving the manuscript. We have addressed all of their comments that appear in red in the revised draft. We hope that you find the manuscript suitable for publication.

Reviewer 1

1. I have no major issues with this manuscript but I should highlight that the

techniques used are outside my area of expertise. I do have several minor points.

Using the terminology “2%, tears; 3%, tears” etc. in table 2 and figures 5 and 7 is

confusing. Having tears after the comma is redundant. Why not replace with

saturation, e.g. 2% saturation. I think this would make the table and figures

easier to follow.

We replaced % tears with % saturated tear lipids

2. In the conversion to pdf it appears that p has be converted to 6 throughout the

discussion.

This was beyond our control. We will make sure all of the conversions are correct in the proofs should the manuscript be accepted for publication.

3. Minor typos/suggestions:

Line 52: lipids, which

Line 56: define HSCT

Lines 94-104: several spaces between words missing

Line 96 and figure 2: there is no b in the figure, please indicate which spectrum is

which.

Line 114: In the figure 4 caption, should you be referring to Figure 3?

Line 120: Use the same nomenclature in the text and Table 1. Also show

significance in table 1.

Line 168: There is no image for Figure 6 a).

Line 207: Tear lipids need to be ordered enough …

Lines 229-230: native TFL was 35% greater than native meibum…

Line 236: between tear and meibum lipids ….

Line 240: As native TL order was 35% greater than native meibum ….

Lines 257-258: Lipid order and pmax were greater in TL than meibum collected

…..

Line 260: similar to ML …

Line 288: repeated using CHCl3

All of the above typos and suggestions were addressed in the revised manuscript. We double checked the draft we submitted and found that between submission and the draft you received, the figures were placed in the wrong places and many of the symbols were not converted correctly.

Reviewer 2

3. the references are not sequentially numbered, redundant statements are made, figure legends are incomplete, etc.

The references are now sequentially numbered

4. Lines 51-52- This is another trivial statement that could be made on science outreach activities to non-specialized audience, but not in a scientific paper, I mean compare order and fluidity with butter and olive oil.

Sorry. We should always be aware of the audience. Most of the readers of our publications in Ophthalmic journals are clinicians and we needed to explain many of the scientific concepts in simple terms. As most of the readers of IJMS will be scientists, we eliminated many of these simplistic statements.

5.  Line 56 – Define HSCT. This can be immediately found upon search, but should be described in the text.

 Thank you. HSCT was spelled out in the two places it was used.

6.    Line 79 – Average NMR spectra were typical of that collected on a 700 MHz machine???

The sentence was correct to: The 1H NMR spectrum of human tear lipid was typical of that collected on a 700 mHz spectrometer (Figure 1). 

7. Lines 92/93- Trivial- There are a large number on CH2 groups in the lipid, so these are the major NMR resonances…

We agree that this sentence is trivial and was removed.

 8. Line 99/100 - Although mentioned, the sigmoidal equation should be described.

The equation is now described in the Methods.

9.  Lines 99/100 – The transition is not fitted to the equation. Equations are fitted to the data.

Thank you. The sentence was corrected to:

Lipid phase transitions were quantified by fitting the data to a two-state, sigmoidal equation using Sigma plot 10 software (Systat Software, Inc., Chicago IL) as follows:

10. Line 114- Complete Figure Legend.

Thank you. The revised legend is:

Figure 3. Lipid phase transitions of tear lipids from a 63 year-old Caucasian male with no dry eye. The CH2 symmetric stretching frequency is related to lipid structural order. The higher the value for the frequency the more disordered the lipid. Symbols are different trials. Solid line is the fit of the data to a sigmoidal equation (See Methods).

 11. Line 120 – Describe how the thermodynamic data was obtained.

A paragraph was added to section 4.5.

Round  2

Reviewer 2 Report

This revised version was significantly improved